# Analysis of Supply and Demand to Enhance Educational Tourism Experience in the Smart Park of Yogyakarta, Indonesia

**Ani Wijayanti [1],\* , Janianton Damanik [2], Chafid Fandeli [3] and Sudarmadji [4]**

[1]  Graduate School of Tourism, Gadjah Mada University, Yogyakarta 55281, Indonesia
[2]  Social and Political Sciences Department, Gadjah Mada University, Yogyakarta 55281, Indonesia; antondmk@yahoo.com
[3]  Forestry Department, Gadjah Mada University, Yogyakarta 55281, Indonesia; chafidfandeli@gmail.com
[4]  Geography Department, Gadjah Mada University, Yogyakarta 55281, Indonesia; sudarmadji@ugm.ac.id
\*  Correspondence: ani.awi@bsi.ac.id; Tel.: +62-81-804-112254

**Abstract:** The Smart Park (also known as Taman Pintar) is a major educational tourist destination in Yogyakarta, which offers a variety of attractions that are very interesting for tourists. The main purpose of tourists visiting Smart Park is to obtain an educational tourism experience. This subjective experience raises specific challenges for Smart Park as it works towards being a competitive destination. The purpose of this study is to analyze the aspects of the educational tourism experience that are affected by tourism demand and supply. Data were collected from surveys that were sent to 150 respondents and were analyzed using path analysis. The results show that tourism demand and supply contributed to the variation of tourism activities by 45.1%, while the remaining was explained by other variables, such as national budget, local budget, ticket sale, and cooperation with some stakeholders. Tourism supply had a higher effect than tourism demand. Tourism demand did not particularly affect tourism experience. However, the results of the path analysis indicate that tourism supply had direct and indirect effects on tourism experience through the variation of tourism activities, with the indirect effect being the most predominant. In the management of Smart Park, there is still a gap between tourism demand and supply, so the tourism experience has not been maximized to its full potential.

**Keywords:** educational tourism; tourism supply and demand; experience; tourism activities

**JEL Classification:** C12; P46; Z32

## 1. Introduction

Tourism is a temporary movement of people to a destination outside of their residence to carry out activities during their stay in the destination, which also requires the preparation of facilities to meet their needs (Pitana and Gayatri 2005). Tourism education is one type of tourism that is mostly found in the city of Yogyakarta. Educational tourism activities vary, ranging from the recognition of schools, customs, and language learning, to seminar and research activities (Wang and Li 2008). The purpose of educational tourism is to recognize education and research, so schools, colleges, and historical sites are often chosen as destinations (Wang and Li 2008). In the world of education, tourism is closely related to academic subjects, such as geography, economics, history, language, psychology, marketing, business, and law. The integration of a number of subjects with tourism studies is essential in enhancing the understanding of tourism and its scope (PSHE 2013).

Educational tourism is one of the most popular tourism businesses in Yogyakarta as a cultural and educational city. The data collected from the Central Bureau of Statistics in a specific region of Yogyakarta show that the number of tourists increased by 17.90% from 2013 to 2014 (3,346,180 people). From the 2013 statistical data, the number of tourists has almost doubled during the last five years (Central Bureau of Statistics (2013)). Educational tours serve as a means of improving academic standards (Smith 2013) so that the study tour program becomes a routine agenda and a part of the school curriculum or an extracurricular activity. The 'Educational Tour' is a program that combines elements of tourism activities with educational content. This program is packed in the form of extracurricular activities in the form of visits to several attractions.

The motivation of tourists to be educational tourists is categorized in several aspects, namely physical, cultural, social, spiritual, and fantasy (Ritchie 2003). In addition, there are two main factors that affect the motivation of tourists, which are namely the demand and supply of educational tourism itself. Tourism demand consists of travel preparation, movement, accommodation and catering, activities at the destination, purchase and personal needs, as well as recording and preserving impressions (Yoeti 2008). Tourism supply includes natural amenities, historical, cultural, religious, infrastructure, means of access and transport facilities, superstructure, and people's way of life. The suitability of demand and supply affects the realization of an optimal tourist experience, which ultimately impacts the satisfaction of tourists and their desire to return. The management of a tourist destination is said to be successful if they are able to offer tourism supply according to the tourists' demands. In this case, the management of tourist destinations in the city of Yogyakarta face considerable challenges in balancing the demand and supply. The demand for educational tourism itself is too broad and diverse to be satisfied by a destination, which can have negative impacts on tourism products in some cases. For example, the forms of traditional arts have changed, lost their meaning, and are no longer authentic due to mass production as one of the efforts to meet tourists' demands (Timothy and Nyaupane 2009). The tourists' main demand is to obtain an educational tourism experience to enhance the understanding about a number of subjects in school.

Smart Park is one of the famous educational tourism destinations in Yogyakarta. Smart Park is the most comprehensive Science Center in Southeast Asia, because it covers several areas of science, including history, physics, biology, math, and chemistry. Smart Park provides learning rides for preschool to high school students. The park is a center of technology-based science and is built with the concept of integrated regional development, while providing space for expression in a friendly educational atmosphere. Smart Park was built in 2003 on a land of 1.2 hectares. The Park is located in strategic area at Panembahan Senopati Street, Yogyakarta, as shown in Figure 1.

The main purpose of tourists visiting Smart Park is to learn, such as in arts, culture, history, and technology. The experiences are subjective as Smart Park cannot give such experiences, but can only create an environment where tourists can actually have these types of experience. The educational tourism experience in Smart Park is the result of tourist interaction during visiting Smart Park (Parahalad and Ramaswamy 2004). The experience itself is manifested because of tourist involvement in various tourist activities (Poulsson and Kale 2004; Echeverri 2005; Brunner-Sperdin and Peters 2009). In this case, Smart Park should understand how to create circumstances that will enhance the experience of tourists (Mossberg 2007).

Given the importance of experience in tourism, the purpose of this study is to find out the extent of educational tourism experience is perceived by tourists through some tourism attractions that are offered by Smart Park Yogyakarta. Tourism can be defined as a complete range of the tourism experience from the departure to the return (TPRG 2003; TPDS 2007). In this case, the experience is the main product that must be managed appropriately by Smart Park to be a competitive destination by designing and providing a memorable tourism experience (Verma et al. 2002). This is one of co-created tourist experience in order to create a sustainable tourist experience.

The co-creation experience is strongly influenced by tourism supply and demand. Larsen (Larsen 2007) conveys the concept of the tourism experience, including expectation, events, and

memories. The expectation in this case is the tourism demand. The events and memories are strongly influenced by the availability of products (i.e., tourism supply). The experience of tourists will be optimized if the manager is able to meet the demand of tourists through the availability of tourist products. Efforts to improve the experience of tourists are strongly influenced by the availability of products and the combination of tourism activities (Dwyer and Kim 2003).

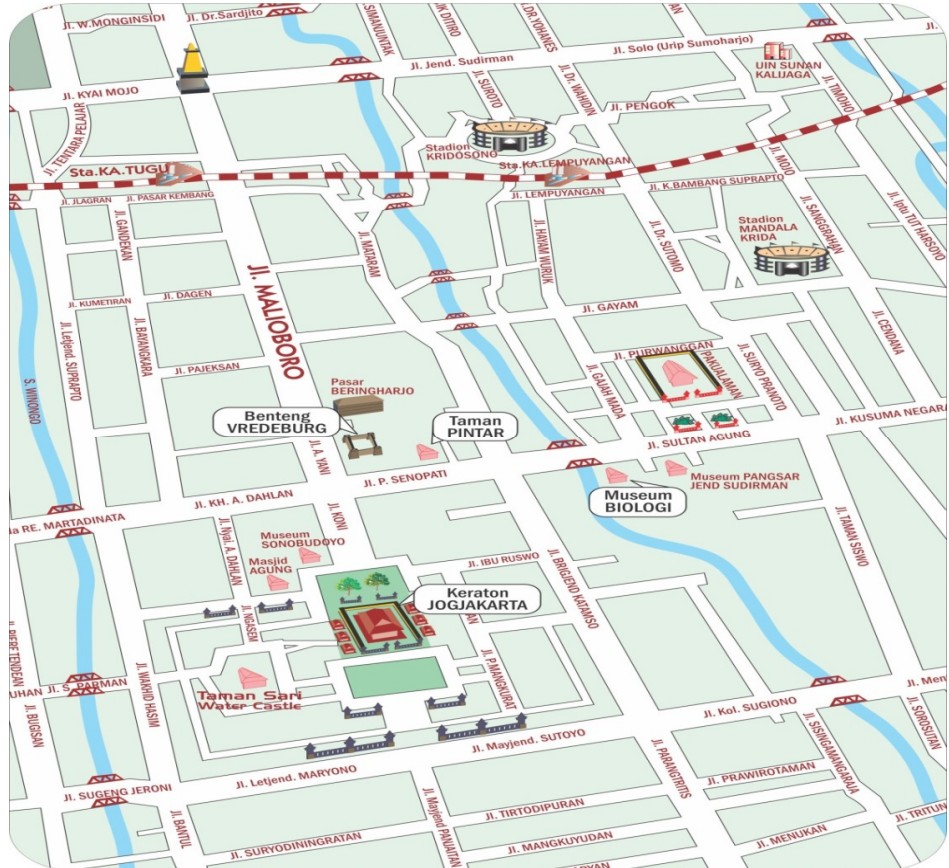

**Figure 1.** Map of Smart Park, Yogyakarta, Indonesia.[1]

## 2. Literature Review

The market share of the educational tourism is divided into several groups of tourists (i.e., adults, elderly, school students and university students) (Ritchie 2003). Meanwhile, Ankomah and Larson (Ankomah and Larson 2002) divided the market share of educational tourism into three categories, which are namely domestic, Europe, and North America.

The products of educational tourism are those that are able to provide an active learning experience as the main objective to be achieved in educational tourism. According to Cohen (Cohen 2008), there are two aspects of education in educational tourism programs, which are the experience and interaction. Tourism products are a mix of different goods and services that are offered as the experience activities for tourists (Cooper and Hall 2008). Educational tourism products have three dimensions, including core products, tangible products, and additional products (Swarbrooke 2002). The core products are those that offer some educational and learning experiences, which are namely the tourism attractions. Tangible products are the core ones, which are packaged into a tour package, while additional products

---

are all additional tangible and intangible services. As one of the educational tourism products, accommodation has relatively simple and inexpensive characteristics, but are able to make social and cultural coordination for tourists, such as home stay (Taylor 2006).

The concept of tourism supply and demand is very useful to create an environment for the development of tourism. Experience becomes the most important factor that can be improved through the provision of attractions, mix of activities, and supporting factors. Supply and demand refer to the ability of a destination to provide social, physical, and economic benefits to the population, as well as a satisfying experience for tourists. The concept of supply and demand is also referred to by other researchers as the concept of attractiveness and competitiveness (Vengesayi 2003). Attractiveness focuses on the demand of tourists and the aspects that attract them to a variety of destinations (Formica 2001), while competitiveness focuses more on the ability of a destination to provide products that can be accepted by tourists, which is often called tourism supply (Kozak and Rimmington 1999). The concept can be seen from two different perspectives. Namely, the attractiveness is seen from the perspective of tourists, while competitiveness is seen from the perspective of the tourist destination (Buhalis 2000). A combination of supply and demand can increase the popularity of a destination. Tourism supply, as an element of competitiveness, refers to the ability of a destination to present a more satisfying tourism experience as compared to other destinations (Hassan 2000).

Over the last two decades, the combination of tourism and information and communication technologies (ICT) has led to considerable changes in tourists' behaviour, which has positively contributed to the growth in tourist demand (Ramos and Rodrigues 2013). The tourism demand is a key determinant of business profitability as a very important element in all of the planning activities (Song and Turner 2006). The tourism demand facilitates economic planners in minimizing the risk of making decisions on the future (Frechtling 2001). An approach based on the concept of supply and demand is very appropriate for enhancing the competitive advantage of tourism destinations. This approach is very useful for determining the appropriate comparisons between investments to be made by managers and what customers look for in a destination.

The main interesting element of a destination is attraction. Attraction is the main motivation for tourists to visit a destination and one of the reasons for selecting a destination (Crouch and Ritchie 1999). Attraction is categorized into five major groups, which are namely cultures, nature, events, recreations, and entertainment (Goeldner and Ritchie 2006). Destination managers play an important role in terms of designing tourism attractions using their initiative and creativity. Offering more tourism attractions will result in tourists staying longer and a greater tourist experience being gained.

Every single traveler has a unique and different personal experience, due to travel planning and post-trips differing greatly (Park and Santos 2016). The tourists' experiences are experienced by the environment of the experience, in which personnel, other tourists, physical environment, products/souvenirs, and themes play a major role (Mossberg 2007). Environment as a product of experience is needed by tourists. They need a safe environment with employees focusing on customers and services, as well as closely cooperating with various parties. Each tourism product provides different experiences for each individual. The assessment of the experience can be seen from the uniqueness of the attraction offered. Destinations with great uniqueness attract visitors, so that they want to spend more time to visit. There are different experience scales that are applicable to each destination, including hedonism, refreshment, local culture, meaningfulness, knowledge, involvement, and novelty (Kim et al. 2012).

Tarssanen (Tarssanen 2005) stated that tourism is multisensory in nature, which results in a comprehensive and positive emotional experience to prove the tourists with a sense of personal transformation. The combination of tourists' experiences is developed to be the perceived image, which can be used to determine the ability of destinations to attract visitors (Horrigan 2009). The image itself is an important element for tourists in selecting a tourism destination (Kamenidou et al. 2009).

Tourism is actually a network and tourists are the individuals who operate in an experimental environment. Therefore, the conception of a co-created experience is very appropriate to apply in of the management of a destination (Van der Duim 2007). The concept provided added value for all of the stakeholders in addition to contributing to the uniqueness and originality of a tourism destination because it is difficult to imitate in other places (Berry et al. 2002). In addition to define the elements in the concept of experience, a set of organizational activities is required to support the presentation of emotional characteristics, behaviors, and other relevant experience performance (Stuart and Tax 2004).

Aho (Aho 2001) developed four core elements of the tourism experience, including emotional experience, learning, practical experience, and transformational experience. However, the tourism experience is frequently in a short and non-continuous period (Ritchie and Hudson 2009). Tourists are generally more motivated by the initial experience through strong mental and emotional impacts instead of the physical characteristics of a tourism destination (Oh et al. 2007). Urry (Urry 2002) argued that tourism incorporates two elements, namely landscapes and sensescapes, which involve multiple senses as the important components of tourism experience. Larsen (Larsen 2007) suggested that the concept of tourism experience includes expectations, events and memories. Brunner-Sperdin and Peters (Brunner-Sperdin and Peters 2009) stated that service delivery is critical in shaping the tourism experience.

Sharing positive experiences in social media has a positive effect on travelers, while unsatisfactory experiences are also able to reduce negative perceptions about travel that ultimately improves post-travel evaluation (Kim and Fesenmaier 2015). The experience and satisfaction of tourists are seen as the aspects that serve as the strategic steps in designing tourism products. Satisfaction comes from customer feelings and expectations as compared to reality. In this case, feedback from tourists and tourism service providers are necessary for assessing the tourists' experience and satisfaction. In view of experience, consumers are the focus in product management and are capable of creating the experience (Parahalad and Ramaswamy 2004).

## 3. Method

This is an explanatory study using four variables, including tourism demand ($X1$), tourism supply ($X2$), tourism activities ($Y$), and tourism experience ($Z$). This was conducted by using a path analysis to analyze the effect of tourism demand and supply on tourism experience directly and indirectly through changing the tourism activities in Smart Park, Yogyakarta, Indonesia (Ghozali 2008). Data were collected by distributing questionnaires to 150 respondents. The survey was conducted on tourists visiting Smart Park, who were selected using the accidental sampling technique. The sample is selected to have a minimum age of 12 years. Most of the selected respondents are junior and senior high school students, who are conducting study tours. Data were analyzed using the Statistical Package for the Social Sciences (SPSS) through two stages, which are namely a multiple linear regression analysis and a path analysis. Regression analysis was carried out to examine the effect of tourism demand and supply as independent variables on tourism activity as a dependent variable. The equation of multiple linear regression analysis was formulated as follows: $Y = \beta_{X_1 Y} X_1 + \beta_{X_2 Y} X_2 + \varepsilon_1$. Meanwhile, a path analysis was carried out to test the direct or indirect effect using the comparison of values $\beta_{X_n Z}$ and $\beta_{X_n Y} \cdot \beta_{YZ}$. If $\beta_{X_n Z} > \beta_{X_n Y} \cdot \beta_{YZ}$, the effect was dominantly direct and for $\beta_{X_n Z} < \beta_{X_n Y} \cdot \beta_{YZ}$, the effect was dominantly indirect, where $X$, $Y$ and $Z$ are latent variables; $\beta$ is the path coefficient and $\varepsilon$ is an estimated error.

The diagram of path analysis model constructed in this study can be seen in Figure 2.

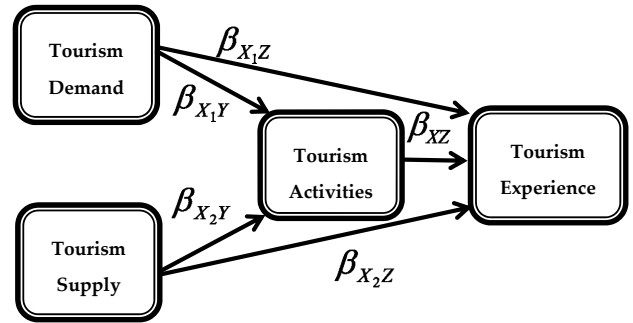

**Figure 2.** Diagram of Path Analysis Model. Source: Redesign Model based on theory of Dwyer and Kim (2003) and Vengesayi (2003).

The measurement instrument is made on a four-point Likert scale, to measure the attitude, opinions, and perceptions of tourists in Smart Park against the four variables as a symptom or phenomenon being measured (Riduwan 2009). A score for the category of answer statements can be seen in Table 1.

**Table 1.** Likert Scale.

| Variable/Score | Choice of Answers | | | |
| --- | --- | --- | --- | --- |
| | **1** | **2** | **3** | **4** |
| Tourism Demand | Strongly Disagree | Disagree | Agree | Strongly Agree |
| Tourism Supply | Strongly Disagree | Disagree | Agree | Strongly Agree |
| Tourism Activities | Strongly Disagree | Disagree | Agree | Strongly Agree |
| Tourism Experience | Strongly Disagree | Disagree | Agree | Strongly Agree |

Source: Riduwan (2009).

## 4. Results and Discussion

### 4.1. Characteristics of Respondents and Description of Variables

Based on the results of the study, most visitors were 12–24 years old (78.7%), with the educational levels of senior high school (40.7%) and junior high school (33.3%), as shown in Table 2.

**Table 2.** Characteristics of respondents.

| No | Description | Frequency (People) | Percentage (%) | No | Description | Frequency (People) | Percentage (%) |
| --- | --- | --- | --- | --- | --- | --- | --- |
| 1 | Gender | | | 4 | Origin | | |
| | Male | 54 | 36.0 | | Yogyakarta | 34 | 22.7 |
| | Female | 96 | 64.0 | | Beyond Yogyakarta, in Java Island | 103 | 68.7 |
| | | | | | Beyond Java Island | 3 | 8.7 |
| 2 | Age | | | 5 | Education | | |
| | 12–40 Th | 118 | 78.7 | | Junior High School | 50 | 33.3 |
| | 25–34Th | 19 | 12.7 | | Senior High School | 61 | 40.7 |
| | 35–44 Th | 6 | 4.0 | | Diploma | 12 | 8.0 |
| | 45–54 Th | 6 | 4.0 | | Bachelor | 21 | 14.0 |
| | 55–64 Th | 1 | 0.7 | | Postgraduate | 1 | 0.7 |
| | | | | | Others | 5 | 3.3 |
| 3 | Marital Status | | | 6 | Frequency of Visits | | |
| | Married | 29 | 19.3 | | 1 time | 72 | 48.0 |
| | | | | | 2–3 times | 62 | 41.3 |
| | Unmarried | 121 | 80.7 | | > 3 times | 16 | 10.7 |

Note: field survey by authors, 2016.

The variables have their descriptions explained in Table 3. This table describes the tourists' perception of the four research variables. The greatest demand from tourists (80%) was to obtain the experience of learning art, culture, and new language. A total of 80% of tourists stated that educational tourism attractions vary widely. Experiences in arts, culture, history, and new technology were demanded by 70%, but some respondents (44.7%) perceived that the level of experience is still low, particularly in language learning. There is still a gap between the demand and supply of educational tourism, which is an important recommendation for the management of Smart Park.

**Table 3.** Variables Description.

| Questionaire Item/Likert Scale | 1 F | 1 % | 2 F | 2 % | 3 F | 3 % | 4 F | 4 % | Questionaire Item/Likert Scale | 1 F | 1 % | 2 F | 2 % | 3 F | 3 % | 4 F | 4 % |
|---|---|---|---|---|---|---|---|---|---|---|---|---|---|---|---|---|---|
| **Tourism Demand (X1)** | | | | | | | | | **Tourism Supply (X2)** | | | | | | | | |
| Souvenirs | 8 | 5.3 | 17 | 11.3 | 779 | 52.6 | 46 | 30.7 | Educational tourism | 7 | 4.7 | 7 | 4.7 | 73 | 48.7 | 63 | 42.9 |
| Transportation | 13 | 8.7 | 31 | 20.7 | 58 | 38.7 | 48 | 32.0 | Transportation | 8 | 5.3 | 28 | 18.7 | 88 | 58.7 | 26 | 17.3 |
| Pilgrimage Activities | 20 | 13.3 | 50 | 33.3 | 64 | 42.7 | 16 | 10.7 | Accomodatiom | 14 | 9.3 | 30 | 20.0 | 77 | 51.3 | 29 | 19.3 |
| Learn and Culture | 10 | 6.7 | 30 | 20.0 | 77 | 51.3 | 33 | 22.0 | Special needs facilities | 6 | 4.0 | 10 | 6.7 | 97 | 64.7 | 37 | 24.7 |
| Conference/meetiing | 3 | 2.0 | 20 | 13.3 | 83 | 55.3 | 44 | 29.3 | Toilet | 10 | 6.7 | 21 | 14.0 | 85 | 56.7 | 34 | 22.7 |
| Learn new language | 4 | 2.7 | 23 | 15.3 | 80 | 53.3 | 43 | 28.7 | Souvenirs | 8 | 5.3 | 23 | 15.3 | 77 | 51.3 | 42 | 28.0 |
| Information Services | 28 | 18.7 | 72 | 48.0 | 39 | 26.0 | 11 | 7.3 | Photographers services | 20 | 13.3 | 46 | 30.7 | 67 | 44.7 | 17 | 11.3 |
| Culinary servuces | 22 | 14.7 | 64 | 42.7 | 50 | 33.3 | 14 | 9.3 | Parking area | 10 | 6.7 | 52 | 34.7 | 72 | 48.0 | 16 | 10.7 |
| Learn new tcehnology | 10 | 6.7 | 29 | 19.3 | 82 | 54.7 | 29 | 19.3 | Culinary services | 7 | 4.7 | 29 | 19.3 | 86 | 57.0 | 28 | 18.7 |
| Photographer services | 24 | 10.0 | 46 | 30.7 | 61 | 40.7 | 19 | 12.7 | Information services | 6 | 4.0 | 22 | 14.7 | 88 | 58.7 | 34 | 22.7 |
| **Tourism Activities (Y)** | | | | | | | | | **Tourism Experience (Z)** | | | | | | | | |
| Pilgrimage activities | 5 | 3.3 | 11 | 7.3 | 91 | 60.7 | 43 | 28.7 | Leraning language | 19 | 12.7 | 64 | 42.7 | 49 | 32.7 | 18 | 12.0 |
| Conference/meeting | 7 | 4.7 | 33 | 22.0 | 86 | 57.3 | 24 | 16.0 | Learning art and culture | 10 | 6.7 | 35 | 23.3 | 73 | 48.7 | 32 | 21.3 |
| Learn history | 19 | 12.7 | 58 | 38.7 | 65 | 43.3 | 8 | 5.3 | Learning new technology | 6 | 4.0 | 37 | 24.7 | 67 | 44.7 | 40 | 26.7 |
| Learn new language | 1 | 0.7 | 33 | 22.0 | 91 | 60.7 | 25 | 16.7 | Learning history | 9 | 6.0 | 45 | 30.0 | 62 | 41.3 | 34 | 22.7 |
| Learn art and culture | 20 | 13.3 | 49 | 32.7 | 68 | 54.3 | 13 | 8.7 | Pilgrimage activities | 26 | 18.0 | 61 | 40.7 | 44 | 29.3 | 18 | 12.0 |
| Field study/research | 2 | 1.3 | 16 | 10.7 | 79 | 52.7 | 53 | 35.3 | Involved with local communities | 14 | 9.3 | 45 | 30.0 | 64 | 42.7 | 27 | 18.0 |
| Learn new technology | 2 | 1.3 | 10 | 6.7 | 74 | 59.3 | 64 | 42.7 | | | | | | | | | |

Note: Field survey by authors, 2016.

## 4.2. Instrument Test

Instrument testing was conducted on 30 respondents, including a validity and reliability test. The validity test is conducted by comparing correlated-item total correlation with *r*-table values, which have 30 respondents that are equal to 0.374. The reliability test was conducted by comparing the value of Cronbach Alpha to a critical value of 0.6. Table 4 shows that questionnaires are valid and reliable, since the value of all correlated-item total correlations is greater than 0.374 and the Cronbach Alpha value is greater than 0.6.

**Table 4.** Instrument Test.

| **Validity Test** | | | | | | | |
|---|---|---|---|---|---|---|---|
| Questionaire Item | Correlated-Item Total Correlation | Questionaire Item | Correlated-Item Total Correlation | Questionaire Item | Correlated-Item Total Correlation | Questionaire Item | Correlated-Item Total Correlation |
| X1.1 | 0.651 | X2.1 | 0.610 | Y1 | 0.592 | Z1 | 0.759 |
| X1.2 | 0.458 | X2.2 | 0.546 | Y2 | 0.774 | Z2 | 0.883 |
| X1.3 | 0.791 | X2.3 | 0.758 | Y3 | 0.757 | Z3 | 0.841 |
| X1.4 | 0.512 | X2.4 | 0.636 | Y4 | 0.678 | Z4 | 0,724 |
| X1.5 | 0.386 | X2.5 | 0.829 | Y5 | 0.460 | Z5 | 0.666 |
| X1.6 | 0.791 | X2.6 | 0.755 | Y6 | 0.783 | Z6 | 0.594 |
| X1.7 | 0.486 | X2.7 | 0.787 | Y7 | 0.418 | | |
| X1.8 | 0.508 | X2.8 | 0.780 | | | | |
| X1.9 | 0.548 | X2.9 | 0.861 | | | | |
| X1.10 | 0.791 | X2.10 | 0.612 | | | | |
| **Reliability Test** | | | | | | | |
| Reseach Variable | | | | Cronbach Alpha | | | |
| X1 | | | | 0.743 | | | |
| X2 | | | | 0.771 | | | |
| Y | | | | 0.756 | | | |
| Z | | | | 0.785 | | | |

Note: Field survey by authors, 2016.

### 4.3. Normality Data Test

The normality data test is conducted to understand the distribution of normal data, which is required for further data analysis. Figure 3 shows that the data were normally distributed. In the Normal P-P Plot, the data is spread around the diagonal line and follows the direction of the diagonal line, so that the regression model meets the assumption of normality data. In the scatter plot, the data spread evenly and did not form a certain pattern, so that the data were assumed to be normal and feasible for data analysis.

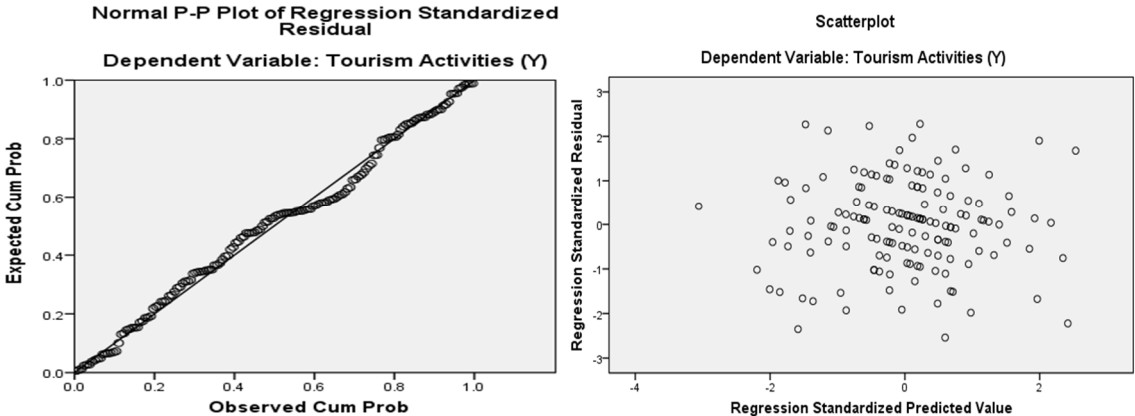

**Figure 3.** Normal PP-Plot and scatter plot. Source: field survey by authors, 2016.

### 4.4. Data Analysis

Data analysis using SPSS was performed in two stages, which are namely a regression analysis and a path analysis. The results of a multiple linear regression analysis can be seen in Table 5.

**Table 5.** Coefficients [a] (Multiple Linear Regression Analysis).

| Model | Unstandardized Coefficients | | Standardized Coefficients | t (Partial Effect) | Significance |
|---|---|---|---|---|---|
| | B (Regression Coefficient) | Standard Error | Beta | | |
| (Constant) | 13.961 | 1.693 | | 8.249 | 0.000 |
| Tourism Demand (X1) | −0.045 | 0.058 | −0.066 | −0.780 | 0.437 |
| Tourism Supply (X2) | 0.263 | 0.054 | 0.411 | 4.855 | 0.000 |

[a] Dependent variable: Tourism Activities (Y). Source: field survey by authors, 2016.

The linear regression equation was formulated as follows: $Y = -0.66X1 + 0.411X2 + E$. This equation means that an increase in tourism demand by one unit will reduce the tourism activities by 0.66. Meanwhile, an increase in tourism demand by one unit will reduce the tourism activities by 0.411. Overall, it can be interpreted that the tourism demand did not partially affect the choice of tourist activities. The effect of tourism demand and supply can be seen in the significance values of t-statistic and f-statistic. The results of the analysis show that the tourism demand in Smart Park did not partially affect tourism activities (*t*-value = 0.437), while tourism supply significantly affected tourism activities (*t*-value < 0.05; Table 1). However, the tourism demand and supply simultaneously affected the tourism activities in a significant way (*t*-value < 0.95; Table 6).

**Table 6.** Anova [b] (Analysis of Variance).

| Model | Sum of Squares | Df (Degree of Freedom) | Mean Square | F (Simultaneous Effect) | Significance |
|---|---|---|---|---|---|
| Regression | 200.449 | 2 | 100.225 | 12.911 | 0.000 [a] |
| Residual | 1141.124 | 147 | 7.763 | | |
| Total | 1341.573 | 149 | | | |

[a] Predictors: (Constant), Tourism Supply (*X2*), Tourism Demand (*X1*); [b] Dependent Variable: Tourism Activities (*Y*).
Source: field survey by authors, 2016.

The results of the study showed that tourists in Smart Park had a very high level of demand for the tourism attractions that provide arts, culture, and language experiences (80%). However, the experience of learning a new language was demanded by less than 50% of tourists. Smart Park has not designed and provided a tourism experience to be the main product to achieve a competitive destination (Verma et al. 2002). The concept of experience is very important, because it is able to create the unique tourism products that are difficult to imitate by other destinations. The management of Smart Park has designed various educational tourism activities using the product supply as the main element, while tourism demand still plays no strong role. A combination of tourism supply and demand is required in the management of destinations because it can increase the popularity of a destination (Hassan 2000). The ability of tourism demand and supply to explain tourism activity can be seen from the R-value in the model summary table (Table 7).

**Table 7.** Model summary.

| Model | R (Determinant Coefficient) | R Square | Adjusted R Square | Standard Error of the Estimate |
|---|---|---|---|---|
| | 0.451 [a] | 203 | 192 | 3.40011 |

[a] Predictors: (Constant), Tourism Activities (*Y*), Tourism Demand (*X1*). Source: field survey by authors, 2016.

The *R*-value was 0.451, which means that the choice of tourism activities was partially determined (45.1%) by variables of tourism demand and supply, while the remaining of 54.9% was explained by other variables that were not examined in the study. From the results of the survey, other variables that affected the variation of tourism activities were budgets, including the national and local budgets, ticket sale, room rental, government subsidies, and cooperation with other parties. Smart Park has collaborated with PT Sarihusada Generasi Mahardhika (PT SGM) in developing several areas. It has also collaborated with educational institutions in developing tourism attractions, especially a puppet stage attraction that is performed by students of Sanata Dharma University Yogyakarta using an English dialogue. The path analysis was conducted to determine the direct or indirect effect of tourism supply on tourism activities, the results of which are shown in Table 8.

**Table 8.** Coefficients [a] (Path Analysis).

| Model | Unstandardized Coefficients | | Standardized Coefficients | t (Partial Effect) | Significance |
|---|---|---|---|---|---|
| | B (Regression Coefficient) | Standard Error | Beta | | |
| (Constant) | 3.224 | 2.202 | | 1.464 | 0.145 |
| Tourism Supply (*X2*) | 0.095 | 0.064 | 0.117 | 1.471 | 0.143 |
| Tourism Activities (*Y*) | 0.495 | 0.100 | 0.393 | 4.932 | 0.000 |

[a] Dependent Variable: Tourism Experience (*Z*). Source: field survey by authors, 2016.

Table 8 shows the path analysis of the three variables, which are namely tourism supply, tourism activities, and tourism experiences. The tourism demand variable is not involved because the coefficient value in the first regression process was a negative number of −0.045. The results of the analysis in Table 8 are used as a reference in creating path diagrams and calculating direct and indirect effects. The path diagram of the analysis results can be seen in Figure 4.

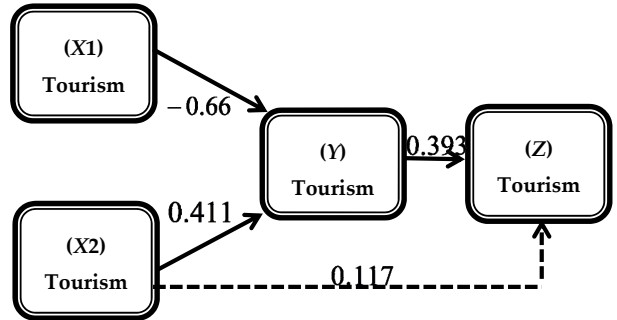

**Figure 4.** Diagram of Path Analysis. Result of data analysis using SPSS.

The diagram of the path analysis model in Figure 4 shows that the tourism supply directly and indirectly affected the tourism experience in Smart Park. The indirect effect was more dominant than the direct one, as shown by the following calculation.

$$\beta_{X_2Z} = 0.117$$

$$\beta_{X_2Y} \cdot \beta_{YZ} = 0.411 \times 0.393 = 0.161$$

$$\beta_{X_2Z} < \beta_{X_2Y} \cdot \beta_{YZ} = 0.117 < 0.161 \ (dominant\ indirect\ effect).$$

The tourism supply affects the tourist experience either directly or indirectly through influencing the variety of tourism activities that are offered by Smart Park. However, the indirect influence is more dominant than direct influence. This shows that changing the tourism activity is able to enhance the environment for cultivating tourist education experiences for tourists. As the largest science center in Southeast Asia, Smart Park is able to provide an extensive tourist experience with regards to learning new technologies, as perceived by 71.4% of the respondents (Table 3).

This case study in Smart Park is consistent with the theory presented by Vengesayi (2003), in which demand and supply were found to simultaneously enhance tourism experiences through educational tourism objects that are designed by the management. From Figure 4, it is seen that tourism demand does not affect the experience of tourists either directly or indirectly through changing the tourism activities in the Smart Park. The experience of tourists is influenced more by the products available in Smart Park, which shows that the demand of tourists is not a major consideration in designing the product. Table 3 shows that over 60% of tourists have high levels of tourism experience on tourism activities related to learning new technologies, arts, and culture, as well as history. Meanwhile, there is still no opportunity to learn new languages in this tourist experience, despite there being a high demand by 70% of the respondents. This case demonstrates the importance of tourism demand to the management. Thus, they will optimize the experience environment and encourage repeat visits.

Nowadays, tourism is an important part of determining individuals' quality of life (Csikszentmihalyi and Hunter 2003). Tourism has become a culture for tourists to freely express the search for unique experiences (Binkhorst 2005a, 2005b). In this case, focusing on the tourism experience was the right strategy applied by Smart Park in the management of tourism products, because an experience can touch the heart of tourists when compared to products or services (Pine and Gilmore 1999). Creating and supplying the tourism experience is important for Smart Park to be able to survive in an increasingly competitive environment in the future. The products designed by the management of Smart Park should be able to optimize the stay of tourists. One of the efforts to maximize tourism experience is to create an effective service by improving tourism resources. In this case, the engagement of stakeholders is essential for the development of tourism in a positive and beneficial way (Mitchell 2001).

## 5. Conclusions

The most dominant demand for the educational tourism experience in Smart Park Yogyakarta was to learn arts, culture, language, history, and new technology. Currently, the tourism experience is lacking the opportunity to learn new languages. Learning languages is one of the main motivations in educational tourism (Ritchie 2003; Cohen 2008). In this case, the Taman Pintar managers face challenges in developing educational tourism products that are related to language learning. The management system in Taman Pintar prioritizes the availability of funds, so that the demand for tourism has not become a major consideration.

As mentioned previously, there is no opportunity to learn languages despite tourists' demands. From the model constructed, it can be concluded that the tourism demand did not partially affect tourism experience either directly or indirectly. However, the tourism supply had a greater influence on the variation of tourism activities, which occurs in an indirect manner. In the management of educational tourism in Smart Park, there is still a gap between tourism demand and supply, which means that the tourism experience has not been fully optimized. Experience becomes the most important factor that can be improved by changing the availability of attractions, mixing of activities, and other supporting factors (Dwyer and Kim 2003). Experience is an important factor because it can improve the competitiveness of Taman Pintar itself (Hassan 2000). The management can minimize the gap between demand and supply by conducting market research. Market research is conducted through cooperation with educational institutions to determine the purpose of the study tours that are part of the curriculum. In addition to conducting market research, management can also offer products to educational institutions as a way to introduce the tourism products owned. In this case, the institution needs to have a clear picture and assist in preparing the tour study agenda.

Limitations of the study. The analysis of the educational tourism experience in this study involved only two independent variables, which are namely tourism demand and supply. The ability of both variables to explain the change in tourism activities was less than 50%. Other variables affecting tourism activities are different types of budget and cooperation with stakeholders. Both of the variables can be recommended for further investigation in future studies.

Contribution of the study. The results of this study show that the tourism experience in Smart Park is still not optimal, because there is still a gap between tourism demand and supply in the management of Smart Park. In terms of demand, there are still aspects of tourism experiences that are not fulfilled, such as the desire to learn new languages. This forms recommendations for the management of Smart Park, particularly encouraging them to design tourism attractions that can enhance the learning of new languages.

**Author Contributions:** A.W. conducted a thorough research, including data collection, data analysis up to the article preparation. J.D. gives guidance in the writing of background and interpretation of data analysis results. C.F. and S. provide guidance on writing techniques and the literature review.

**Conflicts of Interest:** The authors declare no conflict of interest.

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
