# Peer review of "Analysis of Supply and Demand to Enhance Educational Tourism Experience in the Smart Park of Yogyakarta, Indonesia"

_economies, doi:10.3390/economies5040042_

Round 1

Reviewer 1 Report

Grammar and Syntax

Better proof-read and edit needed to reduce some of the syntax & grammar errors; as well as enhance readability and clarity for the reader. 

Introduction

Some of the cited literature and resources are a little dated (e.g. Jafar & Ritchie, 1981;  Pitana & Gayatri, 2005) - recommend that authors critically discuss and review some of the more current literature related to the topic (e.g: McKercher & Tung, 2015; Crouch & Perdue, 2014, etc). It will serve to update some of the recent research and disciplinary focus on tourism studies. 

Further, Figure 1 (Jafar & Ritchie, 1981) seems to be a scanned diagram - ensure that IP has been sought and approved from original authors. 

There seems to be the interchange of terminology in terms of educational tourism, tourist experiences and visitor learning experiences. There can perhaps be clearer definition and articulation and/or focus on explaining educational tourism and its core components - as per indicated by the authors. For example, does the visitor educational and learning experience relate to interpretation as well? How about the notion of co-creation as briefly discussed by the authors? Additionally, perhaps there can be explanation of whether the emphasis is on formal or information learning aspects of educational tourism?

Literature Review

As per the comments above on the academic literature cited - there should be the inclusion and critique of more current literature and resources.  Most of it is from the 2000s and earlier - scholarship and research into tourism education has progressed and developed significantly since then. For example regarding tourist experiences, there has been numerous more recent studies on the components of the tourist experience - e.g. Park & Santos, 2016, Kim, Ritchie & McCormick, 2012, Su, Lebrun, Bouchet, et al. (2016) and etc.... 

The discussion of the concepts relating to tourism supply and demand should be better discussed; and more clearly articulated within the context of educational tourism - presently, it seems to focus more on the tourist experience and destination management considerations. 

Method

Better edit and proof read needed - several errors observed and there seems to be some incomplete sentences? (E.g. lines 170 - 173). Figure 3 - I am not grasping the dimensions in the path analysis model (it is not in English? and / or not explained in the preceding paragraph? - non-English labels / variables / dimensions utilized should be translated. Your topic focus was to be on tourism education - you need to be able to clearly demonstrate the linkages of the dimensions of tourism demand, supply, activities and experiences towards that as per your research objectives. 

Also further explain your data collection and sampling method; as well as how the 150 respondents were recruited for the study - this was not explained. 

Results & Discussion / Conclusion

As per the observations above, there can be a clearer articulation of the implications for tourism education - most of the discussion was focused on the dimensions related to tourism experiences and demand vis-a-vis supply. Further explain its implications on visitor educational / learning outcomes and the notion of value co-creation (as per introduced in the earlier sections). Provide further recommendations for destination / site management on how they may be able to reduce the demand-supply gap indicated to enhance the educational tourism visitor outcomes. 

Author Response

Grammar and Syntax

Better proof-read and edit needed to reduce some of the syntax & grammar errors; as well as enhance readability and clarity for the reader. 

Yes, we have checked the language of our manuscript generally.

Introduction

Some of the cited literature and resources are a little dated (e.g. Jafar & Ritchie, 1981;  Pitana & Gayatri, 2005) - recommend that authors critically discuss and review some of the more current literature related to the topic (e.g: McKercher & Tung, 2015; Crouch & Perdue, 2014, etc). It will serve to update some of the recent research and disciplinary focus on tourism studies. 

Yes, we added new references, there are; McKercher & Prideaux (2014) and Crouch & Perdue (2014).

Further, Figure 1 (Jafar & Ritchie, 1981) seems to be a scanned diagram - ensure that IP has been sought and approved from original authors. 

I haven’t chance to contact the author and I decided to drop off the diagram.

There seems to be the interchange of terminology in terms of educational tourism, tourist experiences and visitor learning experiences. There can perhaps be clearer definition and articulation and/or focus on explaining educational tourism and its core components - as per indicated by the authors. For example, does the visitor educational and learning experience relate to interpretation as well? How about the notion of co-creation as briefly discussed by the authors? Additionally, perhaps there can be explanation of whether the emphasis is on formal or information learning aspects of educational tourism?

Some revisions have done related to the term educational tourism that can be seen in paragraph 2. Meanwhile, on the terms of experience, has been added explanations that can be seen in paragraph 6. While co-creation itself is already in paragraph 7 and clarified in paragraph 8.

Literature Review

As per the comments above on the academic literature cited - there should be the inclusion and critique of more current literature and resources.  Most of it is from the 2000s and earlier - scholarship and research into tourism education has progressed and developed significantly since then. For example regarding tourist experiences, there has been numerous more recent studies on the components of the tourist experience - e.g. Park & Santos, 2016, Kim, Ritchie & McCormick, 2012, Su, Lebrun, Bouchet, et al. (2016) and etc.... 

The discussion of the concepts relating to tourism supply and demand should be better discussed; and more clearly articulated within the context of educational tourism - presently, it seems to focus more on the tourist experience and destination management considerations. 

Yes, we added new references.

Some literature have been added to enrich the literature sources, including Park & Santos (2016) and Kim, Richie & McCormick ()2012. The concept of supply and demand has been described in paragraph 4 of the literature review section and supplemented with the description in paragraph 5.

Method

Better edit and proof read needed - several errors observed and there seems to be some incomplete sentences? (E.g. lines 170 - 173). Figure 3 - I am not grasping the dimensions in the path analysis model (it is not in English? and / or not explained in the preceding paragraph? - non-English labels / variables / dimensions utilized should be translated. Your topic focus was to be on tourism education - you need to be able to clearly demonstrate the linkages of the dimensions of tourism demand, supply, activities and experiences towards that as per your research objectives. 

Also further explain your data collection and sampling method; as well as how the 150 respondents were recruited for the study - this was not explained. 

The methodology has been revised, consist of;

1.      Proof read have been done generally

2.      Incomplete sentences have been revised

3.      Variables in path analysis have been translated in English

4.      The variable dimensions are detailed in Table 3 (variable Description) along with survey results of 150 respondents.

Results & Discussion / Conclusion

As per the observations above, there can be a clearer articulation of the implications for tourism education - most of the discussion was focused on the dimensions related to tourism experiences and demand vis-a-vis supply. Further explain its implications on visitor educational / learning outcomes and the notion of value co-creation (as per introduced in the earlier sections). Provide further recommendations for destination / site management on how they may be able to reduce the demand-supply gap indicated to enhance the educational tourism visitor outcomes. 

The results and discussion section have been revised.

The section has been added by instrument and normality data test. The influence of educational tourism visits to the tourist experience is seen from the results of path analysis. The analysis explains the direct and indirect effects between variables. Recommendations to minimize the gap between tourism demand and supply are explained in the conclusions section. Additionally, research limitations and contributions have also been added to the end of the article.

Reviewer 2 Report

The journal article "Analysis of Supply and Demand to Enhance Educational Tourism Experience in The Smart Park of Yogyakarta, Indonesia" attempts to analyse educational experiences of visitors to Taman Pintar, however it fails to fully answer its objectives. The article is not clean and requires a comprehensive edit. The introduction needs a section on the The Smart Park of Yogyakarta, to give the reader a sense of place. A case study description and location map would be sufficient.

The literature review would benefit from more current references and an improved link between the major theoretical concepts of Educational Tourism and supply and demand. Some current case studies of similar work in both fields would have improved the literature review.

The methodology needs much improvement. Firstly, there is no description of where/when/how the surveys were conducted. Survey questions need to be outlined or a sample survey attached as an appendix. The accidental sampling method needs more explanation along with the quantitative analysis process and model formulation. The study could have benefited from some qualitative components to reinforce the numbers and statistics provided by the survey data.

In the results section, the characteristics of respondents could be summarised in a table and the results illustrated in Figure 4 were very difficult to read and interpret. Figure 5 and the following calculation were confusing and require greater discussion.

The conclusion is largely descriptive and lacks any theory and should have contained several of the key references from the literature review. What were the key highlights from the research that you want the reader to be aware of? Limitations of study require more thought: Sample size? Time period? Number of stakeholders? Quantitative vs Qualitative? Contribution of study needs to focus on the articles contribution to a specific field of research.

Author Response

The journal article "Analysis of Supply and Demand to Enhance Educational Tourism Experience in The Smart Park of Yogyakarta, Indonesia" attempts to analyse educational experiences of visitors to Taman Pintar, however it fails to fully answer its objectives. The article is not clean and requires a comprehensive edit. The introduction needs a section on the The Smart Park of Yogyakarta, to give the reader a sense of place. A case study description and location map would be sufficient.          

 Yes, the introduction have been revised by adding some descriptions of educational tourism concepts, tourist experiences and supply and demand. Smart Park map has been added to provide a clear picture of the object of research.

The literature review would benefit from more current references and an improved link between the major theoretical concepts of Educational Tourism and supply and demand. Some current case studies of similar work in both fields would have improved the literature review.                         

Several literature have been added to enrich the literature sources, including McKercher & Prideaux (2014); Crouch & Perdue (2014); Park & Santos (2016); Kim, Ritchie & McCormick, 2012.                                                                               

The methodology needs much improvement. Firstly, there is no description of where/when/how the surveys were conducted. Survey questions need to be outlined or a sample survey attached as an appendix. The accidental sampling method needs more explanation along with the quantitative analysis process and model formulation. The study could have benefited from some qualitative components to reinforce the numbers and statistics provided by the survey data.

Yes, the methodology have been improved as suggestion.

The first paragraph in the methodology section has been given additional descriptions of the object of the study, while the study time is seen in the year used as the field data source. The questionnaire question in the survey can be seen in table 3 which is supplemented with survey results. Meanwhile, sampling techniques have also been added in the first paragraph in the research methodology section.

In the results section, the characteristics of respondents could be summarised in a table and the results illustrated in Figure 4 were very difficult to read and interpret. Figure 5 and the following calculation were confusing and require greater discussion.

Yes, revision have been done.

Table characteristics of respondents have been added and presented in table 2. Table line regression analysis and path analysis diagram has been given additional explanation to make it easier to understand the results of the analysis.

The conclusion is largely descriptive and lacks any theory and should have contained several of the key references from the literature review. What were the key highlights from the research that you want the reader to be aware of? Limitations of study require more thought: Sample size? Time period? Number of stakeholders? Quantitative vs Qualitative? Contribution of study needs to focus on the articles contribution to a specific field of research.

Yes, the conclusion have been revised and provided more explanation.

In the conclusion section has added some explanations, including limitations and contributions. As for the sample size has been added in the first paragraph in the methodology section.

Reviewer 3 Report

Noted:
1) Problem statement is clear.
2) Research method did not explain clearly (minor revision)
3) Results and finding are good explanation.
4) The conclusion was clearly stated.

Author Response

Revisions have been made on several points as follows;  1. Abstract  2. English language and style  3. Literature review  4. Reseach methodology  5. Data Analysis

Reviewer 4 Report

Title:

The title is precise with the aim and scope of the journal on the economic sector, especially tourism as an industry, as well as relevant to the journal issue about the tourism economy, concise and specific about educational tourism experience located in The Smart Park of Yogyakarta. 

Abstract:

The abstract has been structured in a structured manner, but the background description is less focused or long enough. There is no short explanation of the method used (not yet in accordance with the abstract structure: background-methods-results-conclusion as in Instructions for Authors).

Originality/Novelty:

The question has authenticity, but it has not been defined clearly and well. This article will contribute positively to the development of tourism education studies, in particular, the management of tourism sites that are integrated with the educational materials and values, ie tourism and learning about the environment that can not be found in educational places.

Significance:

Briefly, the result of the study are interpreted long enough, but the main conclusions are supported by the results.

The quality of Presentation:

The writing of articles has been properly prepared and the review literature provided is sufficiently authoritative and comprehensive to support the data and arguments presented in the research results. Nevertheless, the authors present only the results of the study and do not present a contributory discussion for the development of educational tourism studies from the analysis of future supply and demand.

Scientific Soundness:

The design of the study has been prepared precisely and the method used in the form of exploratory studies. In the method section, there are four variables used, but the authors do not briefly describe the indicators of tourism demand (X1), tourism supply (X2), tourism activities (Y), and tourism experience (Z). The authors did not provide a record of the instrument testing of the questionnaire prepared (the questioner and the analysis were attached as a supplement). The author also gave no detailed reason or description about the characteristics of respondents as much as 150 people. Why from Senior High School and Junior High School? The completeness of the methods, tools, and instruments is important so that other researchers can reproduce for the development of this study.

Interest to the Readers: 

The conclusion of the article can invite the reader further to read the entire article, especially the subject of educational tourism. This article offers the development of tourist attractions that not only aim to find entertainment or fun but also to provide educational values for the tourists.

Overall Merit: 

Implementation of demand and supply analysis on educational tourism themes may encourage other researchers/authors widely to research in the development of this field.

Author Response

(The authors gave the same response as above.)

Round 2

Reviewer 1 Report

The Authors have made an effort to address the concerns of the reviewer(s) and have inserted some current academic literature and conceptual support to the manuscript. However, I feel that there can be better integration between the new (inserted) segments and the previous (existing) segments). Some of the flow was a little disjointed and incoherent. Additionally, the new (added) sections and some of the textual discussions had noticeable syntax and grammatical errors and/or inconsistencies. This affected the readability and clarify for the reader.  Sending to a professional technical editor will help to improve the manuscript. 

Author Response

The Authors have made an effort to address the concerns of the reviewer(s) and have inserted some current academic literature and conceptual support to the manuscript. However, I feel that there can be better integration between the new (inserted) segments and the previous (existing) segments). Some of the flow was a little disjointed and incoherent. Additionally, the new (added) sections and some of the textual discussions had noticeable syntax and grammatical errors and/or inconsistencies. This affected the readability and clarify for the reader.  Sending to a professional technical editor will help to improve the manuscript.

I have made changes to the introduction section especially seen in 1st to 3rd paragraph and conclusion part, I've added a theory review. For improving grammar and syntax generally, I have sent article to MDPI English editing by ID: English-1937.

Please review my article. Hopefully there is a significant improvement and able to publish. Looking forward to hearing from you.

Reviewer 2 Report

After reveiwing the changes made by the authors the following further ammendments are required for this paper to be up to a publishable standard:

1) The new references in the introduction are very broad and not related to the topic of educational tourism.

2) The conclusion is still weak and lacks any real theoretical discussion. There needs to be reference to other research or theory.

It should be noted that the methods and results sections of the paper have been vastly improved and strengthened with new references and greater explanations. However, the introduction and conlcusion still require improvements to be made.

Author Response

After reveiwing the changes made by the authors the following further ammendments are required for this paper to be up to a publishable standard:

1) The new references in the introduction are very broad and not related to the topic of educational tourism.

I have made changes to the less relevant references to the introduction, which is seen in the first to third paragraph.

2) The conclusion is still weak and lacks any real theoretical discussion. There needs to be reference to other research or theory.

In conclusion part,  I have added the theory as a study material.

It should be noted that the methods and results sections of the paper have been vastly improved and strengthened with new references and greater explanations. However, the introduction and conlcusion still require improvements to be made.

I've done improvement for introduction that can be seen in the first to third paragraphs, as well as the conclusions. As for grammar and syntax, overall has been proofread by MDPI English Editing by ID: English-1937.

Round 3

Reviewer 2 Report

Author appears to have addressed reviewers suggestions.